# Assessment of Comprehensive Patient-Reported Outcomes Before and After CPAP Therapy in Obstructive Sleep Apnea

**DOI:** 10.3390/biomedicines13112628

**Published:** 2025-10-27

**Authors:** Adriana Loredana Pintilie, Andreea Zabara Antal, Ruxandra Stirbu, Marius Traian Dragos Marcu, David Toma, Raluca Tiron, Carina Adina Afloarei, Mihai Lucian Zabara, Radu Crisan Dabija

**Affiliations:** 1Clinical Hospital of Pulmonary Diseases Iași, 700116 Iasi, Romania; pintilie.adriana.loredana.mg6.7@gmail.com (A.L.P.); stirbu.ruxandra@email.umfiasi.ro (R.S.); dragos.marcu11@yahoo.com (M.T.D.M.); tomadavidro@gmail.com (D.T.); raluca-teodora.tiron@email.umfiasi.ro (R.T.); carina_adina98@yahoo.com (C.A.A.); radu.dabija@umfiasi.ro (R.C.D.); 2Grigore T. Popa University of Medicine and Pharmacy, 700116 Iasi, Romania; mihai-lucian.zabara@umfiasi.ro; 3Department of General Surgery and Liver Transplantation, St. Spiridon University Hospital Iasi, 700115 Iasi, Romania

**Keywords:** sleep apnea, cognition, excessive daytime sleepiness, depression, anxiety, CPAP

## Abstract

**Background**: Obstructive sleep apnea (OSA) impacts daytime alertness, mood, cognition, and quality of life (QoL). Initial alterations in these patient-reported outcomes (PROs) following CPAP therapy, along with their association with adherence and residual respiratory events, are only partially understood. **Materials and methods**: We conducted a retrospective, observational study from January 2024 to May 2025 involving adult patients with OSA. Standardized assessments were performed at baseline and at six months following the initiation of CPAP: Epworth Sleepiness Scale (ESS), WHOQOL-BREF, MoCA, DASS-21, GAD-7, and PHQ-9. The primary endpoint was the change in Patient-Reported Outcomes (PROs) and cognitive performance. The second was to identify associations between these improvements and the degree of adherence to CPAP therapy and residual AHI. **Results**: Seventy-two patients (median age, 57; 65.3% male) with moderate to severe OSA had a baseline median AHI of 34.5/h, ODI of 35.5/h, and a mean nocturnal SpO_2_ of 92.4%. The initial burden was high: median ESS was 14, indicating excessive daytime sleepiness (EDS), present in 68.9%; median MoCA was 24, with 98.6% scoring below 26; median PHQ-9 was 7; median GAD-7 was 5; and 56.8% and 47.9% scored below 50 in physical and psychological domains of WHOQOL-BREF, respectively. After 6 months, group averages showed improvement: ESS decreased to 8.6 ± 3.7, with a 27.0% residual EDS; PHQ-9 was 7.1 ± 4.5; GAD-7 was 6.2 ± 4.1; and MoCA increased to 25.3 ± 2.7, although 48.6% still showed impairment. WHOQOL-BREF scores improved across domains: physical 58.7 ± 14.2, psychological 61.5 ± 13.6, social 63.2 ± 15.4, and environmental 59.8 ± 14.7, with fewer scores below 50 (physical 23.0%, psychological 18.9%). CPAP adherence was high, with a mean of 87.7% and a median of 95%, predicting a greater ESS reduction (*p* = 0.027) and showing a trend toward improvement in PHQ-9 scores (ρ = 0.218; *p* = 0.066). Residual respiratory indices at 6 months (AHI, ODI, SpO_2_) did not correlate with PRO or cognitive scores at the same time point (all *p* > 0.16), nor with their change scores. **Conclusions**: Over the course of six months, CPAP therapy led to notable improvements in sleepiness, mood, anxiety, cognition, and overall quality of life. Nonetheless, many patients continued to face residual problems, mainly excessive daytime sleepiness (EDS) and cognitive challenges. The positive effects were more closely associated with how well patients adhered to the treatment than with remaining levels of residual AHI or ODI.

## 1. Introduction

Obstructive sleep apnea (OSA) is characterized by recurrent upper airway collapse during sleep, leading to cycles of hypoxia and reoxygenation that affect multiple organ systems and approximately 1 billion adults worldwide [1]. Additionally, OSA detrimentally affects health-related quality of life through symptoms such as daytime sleepiness, fatigue, cognitive impairments, and mood disturbances. The failure to eliminate OSA events significantly impacts quality of life, with patients generally recording lower scores compared to the broader population [2]. Studies confirm that, before diagnosis, several functions are impaired, including cognitive functions; mild impairment is observed in over 55% of cases with moderate to severe OSA [3]. Depressive symptoms are noted in 35% of patients, and 32% experience anxiety [4]. During initial evaluation, excessive daytime sleepiness is a common finding, with 56% of adults exhibiting moderate sleepiness [5]. Pathological fatigue is frequently reported, with 61% of patients diagnosed with moderate to severe OSA screening positive for this condition before treatment [6]. The literature indicates that Continuous Positive Airway Pressure (CPAP) remains the primary therapy, and while it improves outcomes, individual responses vary. Due to these complexities, comprehensive patient-reported outcomes covering sleepiness, mood, cognition, and quality of life are essential for quantifying the real-world impact of CPAP and for identifying profiles at risk of suboptimal symptomatic benefits. Consequently, we conducted a retrospective, observational study to assess changes in a multidomain patient-reported outcomes battery before and after CPAP initiation, and to explore associations with adherence and residual respiratory burden in adults with OSA. The present study was designed with the following objectives: to quantify the baseline burden of excessive daytime sleepiness, cognitive impairment, depression, anxiety, and quality of life impairment in adults with OSA prior to treatment initiation, and to examine baseline correlations between respiratory indices (AHI, ODI, mean nocturnal SpO_2_) and multidomain patient-reported outcomes; to evaluate changes in these patient-reported outcomes following six months of CPAP therapy; to assess the associations between therapeutic adherence and improvement in patient-reported outcomes; and to determine whether residual AHI and CPAP compliance at six months correlate with concurrent patient-reported outcomes or their degree of improvement from baseline. The primary endpoint was the change in Patient-Reported Outcomes (PROs) and cognitive performance. The second one was to identify associations between these improvements and the degree of adherence to CPAP therapy, as well as residual AHI. By addressing these objectives through a comprehensive multidomain assessment, we aim to provide clinicians with evidence-based insights into the real-world symptomatic benefits of CPAP therapy.

## 2. Materials and Methods

### 2.1. Study Design, Participants, and Data Collection

We conducted a retrospective, observational study analyzing medical records of patients diagnosed with sleep apnea syndrome from January 2024 to May 2025 at the Clinical Hospital of Pulmonary Disease Iasi. The inclusion criteria encompassed individuals aged 18 years or older with a confirmed diagnosis of sleep apnea characterized by an AHI of 5 or more events per hour, adherence to CPAP therapy, and no previous diagnosis of OSA or prior treatment with CPAP. Patients were excluded if they were unable to sign or understand the purpose of the study. Incomplete questionnaires were excluded from the final analysis; no imputation was applied. Additionally, subjects with neurological or psychiatric disorders, acute intercurrent conditions, or incomplete questionnaires were also excluded. Out of a total of 159 patients, after evaluating their records, 72 met the eligibility criteria and were included in the study. In accordance with the exclusion criteria, no enrolled participant was under psychiatric treatment or receiving antidepressant medication at baseline. Table 1 outlines the inclusion and exclusion criteria.

The Ethics Committee of the Clinical Hospital of Pulmonary Diseases Iasi approved the study protocol (approval number 129, dated 26 May 2025). All participants provided written informed consent during their initial clinical evaluation. Data were collected during routine medical care and stored in the hospital database before the start of the study. No data were accessed, extracted, or analyzed before obtaining ethical approval; such analysis only began after approval, in accordance with institutional and international ethical standards.

### 2.2. Sleep Test and Multidomain Patient Outcome Evaluation

Diagnosis was made using Polygraph recordings obtained with the SleepDoc Porti 9 device (Löwenstein Medical, Germany, Kronsaalsweg 40, 22525 Hamburg), a type III device, during either a single night of hospitalization or at home. According to the American Academy of Sleep Medicine (AASM) guidelines, sleep apnea was defined as an Apnea-Hypopnea Index (AHI) of ≥5 events per hour, with severity classified as mild (5 ≤ AHI < 15), moderate (15 ≤ AHI < 30), and severe (AHI ≥ 30). Apnea was characterized by a >90% reduction in respiratory flow lasting at least 10 s, while hypopnea involved a 40–50% flow decrease for at least 10 s, along with a >3% drop in oxygen saturation. Data analysis was conducted manually and confirmed by sleep medicine physicians. Each polygraphy was interpreted once by the patients’ treating sleep physicians according to AASM criteria; no parallel second reads were performed with the purpose of inter-rater reliability estimations.

At the initial presentation, all patients underwent a clinical and paraclinical evaluation, with the diagnosis established through nocturnal respiratory polygraphy. Several standardized instruments were applied during the initial medical visit to assess the baseline state of each individual through a questionnaire.

The Epworth Sleepiness Scale (ESS) assesses daytime sleepiness, with scores from 0 to 10 indicating normal levels, ≥11 signifying EDS, 13–15 representing moderate EDS, and 16–24 indicating severe EDS.

Global cognition was assessed using the Montreal Cognitive Assessment–blind version (MoCA). Scores of ≥19 points were considered normal, 16–18 indicated borderline or possible mild cognitive impairment, and ≤15 suggested probable cognitive impairment.

Depression, Anxiety, and Stress were assessed using the DASS-21 scale, which comprises three subscales, each evaluating a specific subset of symptoms. Each subscale contains seven items that are summed and multiplied by two to produce a score ranging from 0 to 42. Depressive symptoms were classified as follows: 0–9 indicating normal, 10–13 mild, 14–20 moderate, 21–27 severe, and scores of 28 or higher representing an extremely severe level. Anxiety symptoms were scored on the DASS-21 subscale and categorized as 0–7 regular, 8–9 mild, 10–14 moderate, 15–19 severe, and scores of 20 or higher indicating an extremely severe level. Perceived stress was evaluated on the corresponding subscale and classified as 0–14 usual, 15–18 mild, 19–25 moderate, 26–33 severe, and scores of 34 or higher denoting an extremely severe level.

Anxiety was measured using the Generalized Anxiety Disorder-7 (GAD-7), which includes seven items assessing nervousness, uncontrollable or excessive worry, trouble relaxing, irritability, and fear. The scores were summed to a total ranging from 0 to 21. Severity bands were categorized as 0–4 for minimal, 5–9 for mild, 10–14 for moderate, and 15–21 for severe.

Depressed mood, sleep, energy, appetite/weight, feelings of failure/guilt, concentration, psychomotor changes, and suicidality are assessed using the Patient Health Questionnaire-9 (PHQ-9). This 9-item questionnaire is scored from 0 to 27. Severity levels are categorized as follows: 0–4, minimal; 5–9, mild; 10–14, moderate; 15–19, moderately severe; and 20–27, severe. A score of 10 or higher indicates major depression.

Life quality was evaluated using the WHOQOL-BREF tool, which encompasses four domain scores: Physical, Psychological, Social Relationships, and Environment. Domain means were transformed using the WHO4-20 metric and then linearly rescaled to a 1–100 scale. Since universal diagnostic cut points have not been established, we interpreted scores relative to the study population, with prespecified categories defined as follows: ≤50 indicating low quality of life, 50–75 indicating intermediate quality of life, and ≥75 indicating preserved quality of life.

Patients were re-evaluated at 6 months after beginning CPAP therapy using the same tools and physiological parameters. Device-reported adherence was extracted at 6 months. Good adherence was defined a priori as ≥4 h/night on ≥70% of nights, although continuous adherence (%) was also analyzed. This enabled a comparison of device-reported adherence, residual apnea-hypopnea index (AHI), and the therapy’s impact on cognitive function, psycho-emotional state, and overall quality of life.

The study’s primary objective was to compare scores at diagnosis with those following CPAP, assessing changes over a 6-month interval. The paper also aimed to evaluate the therapy’s effects on cognitive function, psycho-emotional well-being, and quality of life.

### 2.3. Statistical Analysis

Utilizing IBM SPSS Statistics for Windows, version 26.0 (IBM Corp., Armonk, NY, USA), we conducted a longitudinal analysis to compare baseline and six-month outcomes using paired *t*-tests or Wilcoxon tests. Additionally, we examined associations using Pearson and Spearman correlations. Changes in paired proportions at prespecified clinical cut-offs (ESS ≥ 10, PHQ-9 ≥ 10, GAD-7 ≥ 10, MoCA < 26, WHOQOL-BREF < 50 per domain) were evaluated with McNemar’s test (exact binomial on discordant pairs). Associations were examined using Pearson correlations when both variables were approximately normally distributed and linear, or Spearman’s rho otherwise. Change scores (Δ6-month −baseline) were analyzed using multiple linear regression, adjusting for age, sex, baseline score, CPAP adherence, and residual AHI. *p*-values for pre/post comparisons are reported directly in Table 2 and Table 3.

Clinical polygraphy and questionnaire scoring were performed per patient by a single treating physician (polygraphy) and by self-administered instruments with deterministic scoring (questionnaires). As independent duplicate ratings were not obtained, inter-rater reliability statistics such as the Interclass Correlation Coefficient (ICC) were not applicable to this data set.

## 3. Results

The study included 72 patients with moderate-to-severe OSA who were recommended CPAP. The median age was 57 years, with a mean of 57.1 ± 12.0 years. Among them, 47 (65.3%) were male and 25 (34.7%) were female. At diagnosis, the median AHI was 34.5 events per hour, ODI was 35.5 events per hour, and the median nocturnal SpO_2_ was 92.4%. Figure 1 displays the baseline characteristics of the study cohort.

The median ESS score was 14 points, suggesting moderate to severe daytime sleepiness at diagnosis. Patients reported mild depressive symptoms, with a median PHQ-9 score of 7 points, and mild anxiety, with a median GAD-7 score of 5 points.

The quality-of-life assessment was conducted utilizing the WHOQOL-BREF questionnaire, revealing a median score of 45 points in the Physical domain, indicative of a considerable decline in physical well-being and functional capacity. The Psychological component attained a score of 48 points. The Social component remained relatively preserved, with patients scoring 56 points, suggesting that interpersonal relationships were less affected. Unexpectedly, the Environmental component achieved the highest median score of 59 points, reflecting an overall positive perception of resources and living conditions.

To evaluate cognitive dysfunction, the MoCA score was used, yielding a median score of 24 points, consistent with mild cognitive impairment.

The assessment of affective symptoms was employed, using the DASS–21 score, which revealed a mild to moderate psychological burden. The Depression sub-score had a median of 12 points, indicating mild-to-moderate depressive symptoms. Similarly, regarding anxiety, there was a score of 10 points on the respective subset, revealing a mild to moderate level of anxiety. A median score of 14 points on the respective sub-scale suggested moderate stress.

Thus, at diagnosis, 68.9% (ESS ≥ 10) of patients reported excessive daytime sleepiness. Regarding mood, clinical symptoms were present in 63.5% of the cases (PHQ-9≥10), and 58.6% showed moderate to severe anxiety (GAD-7 ≥ 10). Cognitive assessment revealed a high rate of impairment, with 98.6% of patients scoring below the MoCA cutoff (<26). The quality of life was also significantly affected, with 56.8% of patients scoring low (<50) in the physical domain of the WHOQOL-BREF, 47.9% in the psychological domain, 41.1% in the social domain, and 52.1% in the environmental domain. Based on the DASS-21 subscales, 72.6% of patients had depression scores ≥ 10, corresponding to at least mild depressive symptoms, while 53.4% had anxiety scores ≥ 8, corresponding to at least mild anxiety. Distribution by severity showed that 41.1% had mild depression, 27.4% moderate, and 4.1% severe; for anxiety, 28.8% fell within the moderate range, 16.4% in the severe range, and 2.7% in the extremely severe range.

No statistically significant sex difference was found in baseline respiratory indices, indicating comparable OSA severity between sexes at diagnosis. Female patients tended to have higher affective symptom burden and slightly impaired cognitive function. Additionally, depression scores were significantly higher in females. Females demonstrated a numerically higher affective symptoms burden, with higher PHQ-9 and GAD-7 scores, although only anxiety reached statistical significance (*p* = 0.011). Cognitive function, assessed by MoCA, was slightly lower in females compared to males (*p* = 0.031). In addition, depressive symptoms (DASS-21) were significantly more pronounced in female participants. No sex differences were detected in stress levels. Quality of life scores (WHOQOL-BREF domains) were similar between sexes, although women showed slightly lower scores in the physical and psychological domains.

After six months of treatment, most questionnaires showed significant improvement. The mean Epworth Sleepiness Scale (ESS) score decreased to 8.6 ± 3.7; however, residual excessive daytime sleepiness remained in 27.0% of patients. Depressive symptoms, assessed with the PHQ-9, decreased to an average score of 7.1 ± 4.5, with 28.4% of cases showing moderate to severe depression (PHQ-9 ≥ 10). Anxiety level also reduced, with mean GAD-7 scores of 6.2 ± 4.1, and 22.9% of patients experiencing clinically significant anxiety (GAD-7 ≥ 10). Cognitive performance, evaluated by MoCA, improved to a score of 25.3 ± 2.7. Nonetheless, cognitive impairment, defined by a MoCA score of less than 26, remained significant in 48.6% of the participants. Life quality was improved, with mean scores of 58.7 ± 14.2 in the physical domain, 61.5 ± 13.6 in the psychological domain, 63.2 ± 15.4 in the social domain, and 59.8 ± 14.7 in the environmental domain. The prevalence of low scores (below 50) declined to 23.0% in the physical domain, 18.9% in the psychological domain, and 16.2% in the social domain, as well as in the environmental domain, 21.6% According to DASS-21, mean scores for depression, anxiety, and stress subscales were 8.5 ± 4.0, 5.1 ± 3.9, and 5.3 ± 4.1, respectively. The prevalence of clinically significant depressive symptoms (≥10) was 25.7%, while anxiety (≥8) was present in 16.2% of patients. Although most parameters improved significantly, many patients still suffered from residual excessive daytime sleepiness (27%) and cognitive impairment (48.6%) at 6-month follow-up. Possible reasons include irreversible neurocognitive injury secondary to chronic intermittent hypoxia, undiagnosed and untreated residual sleep disturbances such as inadequate sleep duration, circadian disorder, periodic limb movement, or comorbid insomnia and non-anatomical OSA characteristics, including instability of ventilatory control or low arousal threshold. These factors underscore the multifactorial complexity of residual symptoms in OSA that are unlikely to be ameliorated by CPAP alone.

There is no association of residual AHI/ODI with mood or cognitive outcomes, suggesting that the beneficial impact of CPAP on these symptoms is not simply a reflection of changes in easily measurable respiratory indices. Adherence, symptom severity at baseline, and comorbid conditions may have a greater impact on patient-reported outcomes than residual event counts, which is consistent with the idea that OSA management should be considered within a multidimensional framework.

At the reevaluation, nocturnal respiratory events, including AHI, ODI, and the mean nocturnal SpO_2_, did not show significant correlations with questionnaire scores. For example, AHI showed no correlation with the Epworth (*p* = 0.915), PHQ-9 (*p* = 0.164), GAD-7 (*p* = 0.497), MoCA (*p* = 0.952), or the physical domain of the WHOQOL-BREF (*p* = 0.706). Likewise, ODI was not associated with PHQ-9 (*p* = 0.236), MoCA (*p* = 0.539), or the physical domain of the WHOQOL-BREF (*p* = 0.850). The average nocturnal SpO_2_ showed a weak, non-significant relationship with MoCA (*p* = 0.312) and the psychological domain of WHOQOL-BREF (*p* = 0.271).

Residual AHI at 6 months of therapy was also not significantly correlated with changes in questionnaire scores. Weak associations were observed for PHQ-9 improvement (*p* = 0.395) and MoCA improvement (*p* = 0.295), whereas Epworth (*p* = 0.994), GAD-7 (*p* = 0.393), and WHOQOL-BREF physical domain improvement (*p* = 0.577) showed no meaningful relationship.

Treatment adherence was generally high among the study population. At six months, the average CPAP compliance was 87.7% (SD 17.1), with a median of 95%. These results suggest that most patients used CPAP consistently, providing adequate therapeutic exposure. CPAP compliance was a significant predictor of clinical improvement. Higher compliance was significantly associated with greater reductions in excessive daytime sleepiness (*p* = 0.027) and showed a positive trend with improvements in PHQ-9 scores (*p* = 0.218, *p* = 0.066). No significant associations were found with the GAD-7 (*p* = 0.359) or MoCA (*p* = 0.558). Interestingly, compliance was negatively correlated with improvements in the WHOQOL-BREF physical domain (*p* = 0.033). Univariate changes in patient-reported outcomes are shown in Figure 2.

Both sexes showed a marked reduction in excessive daytime sleepiness, depressive symptoms (PHQ-9), anxiety (GAD-7), scores at follow-up (all *p* > 0.20), indicating that CPAP Intervention can generate comparable clinical improvement across sexes. Interestingly, quality of life exhibited partially sex- dependent patterns. Males reported significantly higher physical WHOQOL scores after 6 months of CPAP compared to women, with similar scores in the social domain. Regarding the environment component, improvements were observed in both sexes. Still, males demonstrated significantly higher post-treatment scores (*p* < 0.001)—overall, males perceived greater improvement in physical and environmental quality-of-life dimensions.

For each patient-reported outcome, we fit logistic models (Improved vs. Not improved) and linear sensitivity analyses on Δ-scores (defined so that positive scores indicate improvement), adjusting for age, sex, baseline score, CPAP adherence (% over 6 months), and residual AHI. Residual AHI did not independently predict improvement for any outcome (all *p* > 0.05). Consistently, higher CPAP adherence showed significant or near-significant associations with improvements in mood and quality-of-life domains in the linear models (e.g., PHQ-9, WHO-Social, WHO-Environment), whereas residual AHI did not. After adjustment, no post-treatment sex differences were observed for ESS, PHQ-9, GAD-7, or MoCA (all *p* > 0.20). In contrast, quality-of-life recovery favored men in the WHOQOL-BREF Physical (*p* = 0.041) and Environment (*p* < 0.001) domains. A more negligible sex-related effect was also observed in the psychological domain (*p* = 0.032), but this did not persist across sensitivity models. Overall, these findings highlight CPAP adherence—rather than residual respiratory burden—as the principal modifiable driver of 6-month patient-reported benefit. Additional materials related to this study are provided in the Appendix A. 

## 4. Discussion

Our study reveals a significant baseline burden across multiple patient-reported domains in adults with moderate to severe OSA, with significant improvements following six months of CPAP therapy, but with residual symptoms. These results underscore the multifaceted nature of OSA’s impact on daily functioning and warrant careful consideration.

Multidomain Patient-Reported Outcomes in OSA: Quality of Life, mood, cognition, and sleepiness.

A thorough quality of life (QoL) evaluation is crucial for patients with OSA, as this condition impacts various physical, psychological, and social domains beyond traditional clinical and polysomnographic assessments. Schipper and colleagues described QoL as the functional effect of illness and its treatment on a patient, as perceived by the patient. OSA diminishes QoL through symptoms such as fatigue, daytime sleepiness, poor sleep quality, concentration difficulties, memory problems, headaches, and mood swings [7,8]. Although various questionnaires have been utilized to evaluate QoL in patients with OSA, we selected the World Health Organization Quality of Life (WHOQOL) instrument due to its broader, more holistic perspective compared with disease-specific tools that focus exclusively on sleep-related symptoms. The WHOQOL assesses four principal domains (physical health, psychological well-being, social relationships, and environmental factors) using a 5-point scale, and domain scores are conventionally transformed to a 0–100 scale. Domains are classified as “low” for scores below 50, “intermediate” for scores between 50 and 57, and “preserved” for scores above 75 [9]. Our baseline findings revealed substantial QoL impairment, with 56.8% of patients scoring below 50 in the physical domain and 47.9% in the psychological domain, rates comparable to those reported by Bjornsdottir et al. (2) in untreated OSA cohorts. Following six months of CPAP therapy, mean physical domain scores improved from 45 to 58.7 ± 14.2, and psychological from 48 to 61.5 ± 13.6, representing clinically meaningful gains that align with previous studies.

CPAP therapy has demonstrated a significant impact on multiple domains of quality of life (QoL) within a cohort of middle-aged, postmenopausal, non-sleepy women suffering from moderate to severe obstructive sleep apnea (OSA). Following 12 weeks of CPAP treatment, participants exhibited notable enhancements in mood state, daytime alertness, and reductions in symptoms of anxiety and depression compared to conservative management. Over the span of three months, CPAP elicited substantially greater improvements than conservative therapy across all five domains of the Quebec Sleep Questionnaire: hypersomnolence, diurnal symptoms, nocturnal symptoms, emotions, and social interactions, thereby indicating its capacity to ameliorate various facets of QoL for this demographic. Moreover, a precise dose–response relationship was observed, whereby increased adherence was correlated with more pronounced improvements in QoL [10]. This association was independently confirmed in our cohort, where higher CPAP compliance predicted greater symptomatic benefit. A comprehensive meta-analysis conducted by Jing J and colleagues evaluated the effects of CPAP in OSA patients, concluding that pressure therapy has a minimal impact on overall Quality of Life measures. Nonetheless, although CPAP therapy does not enhance general QOL scores, it does improve specific domains, particularly in physical aspects and vitality [11]. Our study’s high median adherence (95%) likely contributed to the robust domain-specific improvements observed, underscoring the critical importance of sustained therapeutic exposure in realizing QoL benefits.

Depression and anxiety are common co-occurring conditions in people with obstructive sleep apnea (OSA). Epidemiological studies consistently show a link between OSA and depression [12]. A close relationship exists between this sleep disease and major depressive disorder (MDD), as they share symptoms like low mood, anxiety, restlessness, fatigue, and difficulty concentrating [13]. Therefore, clinicians need to identify the root cause and treat disease-specific symptoms. An extensive cross-sectional study of 18,980 individuals aged 15–100 years found that those with MDD are five times more likely to have OSA compared to the general population [14]. Our cohort demonstrated baseline prevalence rates of clinically significant depression (PHQ-9 ≥ 10) in 63,5% and anxiety (GAD-7 ≥ 10) in 58.6% of patients, substantially higher than the 35% and 32% rates, respectively, reported by Garbarino et al. [4].

Major depressive disorder can be ruled out by using various questionnaires, such as the Patient Health Questionnaire (PHQ-9). Cut points of 5, 10, 15, and 20 indicate mild, moderate, moderately severe, and severe depression, respectively, and are particularly useful for screening depression. They are easy to implement and can provide rapid diagnostic results in primary healthcare settings [15]. The Generalized Anxiety Disorder 7-item scale (GAD-7) assesses subjective anxiety. Scores of 5, 10, and 15 points indicate mild, moderate, and severe anxiety, respectively. The DASS-21 provides separate sub-scales for Depression, Anxiety, and Stress, and responses on all instruments are interpreted in conjunction with patient symptoms and clinical evaluation during primary care visits [16,17].

Some individuals with OSA exhibit depressive symptoms that improve following CPAP therapy [18]. In cases where patients present with both OSA and depressive symptoms, targeted management of both disorders is required [19]. While short-term, placebo-controlled studies (2–3 weeks) have not consistently shown significant improvements in depressive symptoms following continuous positive airway pressure therapy [20,21], most evidence supports treatment’s beneficial effects on mood disturbances in patients with obstructive sleep apnea [22,23,24,25,26,27,28]. Most research has concentrated on middle-aged populations, with limited data for elderly groups. A recent study of 224 elderly patients (≥70 years) with severe OSA (AHI ≥ 30) reported baseline depression in 23.2% and anxiety in 17%, with 66 patients on antidepressants. After three months, patients receiving CPAP showed marked reductions in depressive (*p* < 0.001) and anxiety symptoms (*p* = 0.016) compared to untreated controls [23]. Similarly, in a sample of 228 CPAP-compliant patients, including 33.1% aged 65 or older, the prevalence of depression decreased markedly from 74.6% to 3.9% over three months, alongside significant reductions in apnea–hypopnea index (AHI) and Patient Health Questionnaire-9 scores [27].

Our results occupy a middle position within this spectrum: at six months, the prevalence of moderate-to-severe depression decreased from 63.5% to 28.4%, and clinically significant anxiety from 58.6% to 22.9%, representing substantial but incomplete resolution. CPAP adherence emerged as a significant predictor of mood improvement, with higher compliance showing a trend toward PHQ-9 score reduction (*p* = 0.218, *p* = 0.066). These findings suggest that sustained therapeutic exposure is necessary for maximal psychiatric benefit, though a considerable minority of patients experience persistent symptoms requiring additional intervention. The therapeutic efficacy of CPAP extends beyond objective physiological improvements and is fundamentally influenced by patient acceptance, tolerance, and subjective experience. Even when clinically appropriate, treatment may yield suboptimal patient-reported outcomes if it is not perceived favorably or is poorly tolerated. The discordance between objective health improvements and subjective symptomatic relief observed in our cohort at six months may partially reflect CPAP-associated adverse effects that counterbalance or mask measurable clinical gains. Specifically, although CPAP therapy demonstrably reduces AHI and improves oxygenation, device-related complications, including mask discomfort, claustrophobia, nasal congestion, and skin irritation, can paradoxically perpetuate or exacerbate psychological distress. Anxiety and depressive symptoms may persist or even intensify secondary to the psychological burden of wearing a mask, device dependence, and altered sleep routines. Furthermore, the physical and social implications of long-term CPAP use can adversely impact self-esteem, body image, and intimate relationships, dimensions that are incompletely captured by conventional quality-of-life instruments, yet critically influence treatment preferences and long-term engagement.

Cognitive dysfunction, a clinical syndrome marked by a decline in mental abilities, may or may not include functional impairment and poses a risk of progressing to dementia. In OSA, cognitive impairment is a standard feature, mainly affecting attention, vigilance, verbal and visual memory, visuo-spatial and constructional skills, and executive functions. At the neuronal level, intermittent hypoxia (IH) caused by OSA induces oxidative stress, endoplasmic reticulum stress, iron accumulation, and neuroinflammation. These processes lead to synaptic dysfunction, gliosis, apoptosis, and impaired neurogenesis, all of which contribute to deficits in learning, long-term cognitive function, and memory [29,30]. Pase et al. studied the link between sleep, OSA, and cognition in the Sleep and Dementia Consortium, using data from 5 US cohorts with 5946 adults (31.5% women; mean age 58–89) with no stroke or dementia. Results showed sleep maintenance correlated positively with cognition, while wakefulness after sleep onset (WASO) correlated negatively. More than five events per hour was associated with lower scores, with similar findings for moderate-to-severe OSA [31]. Cognition can be assessed through patient-reported questionnaires that capture everyday memory and cognitive function, including the Montreal Cognitive Assessment (MoCA), a 30-item scale. Scores below 26 may suggest probable mild cognitive impairment, observed in 47.9% of patients diagnosed with sleep apnea. This percentage rises to over 55.3% in patients with moderate and severe OSA [31].

Our baseline cognitive assessment revealed a high prevalence of impairment (98.6% scoring MoCA < 26), exceeding published rates in comparable OSA cohorts [3,32]. This discrepancy warrants consideration: it may reflect the use of MoCA-Blind rather than the standard MoCA, population-specific factors, including educational attainment and cultural context in our Romanian cohort, or genuine severity of cognitive burden in our tertiary referral population. Following six months of CPAP therapy, mean MoCA scores improved from 24 to 25.3 ± 2.7, yet 48.6% of patients continued to exhibit impairment. This may highlight the incomplete reversibility of OSA-induced cognitive deficits. Even when cognitive scores fail to normalize, CPAP may provide neuroprotective benefits by preventing progressive decline.

A study by D’Rozario et al. examined the effect of CPAP on spindles and cognition in 90 participants (average age 50.6  ±  12.1 years, 72% male, AHI 34.8  ±  24.7 events per hour). Tests of attention, visuo-spatial scanning, executive function, working memory, and vigilance were conducted at the first check-up and after 6 months of CPAP use. The study found that six months of therapy increased spindle density during N2 sleep and improved executive function and working memory [33]. In a retrospective analysis of 171 patients with cognitive impairment and diagnosed OSA (mean age 69.8 ± 10.6 years; 66% male), good adherence over 2–12 months was linked to better cognitive outcomes. Patients with good adherence showed higher follow-up MoCA scores and a 1.2-point increase in MMSE scores compared to those with poor adherence or no CPAP use (*p* = 0.01) [34].

Excessive Daytime Sleepiness (EDS) affects about 40.5–58% of those with OSA, with residual EDS present in 9–22% of patients on CPAP. It impacts safety, productivity, mood, cognition, and overall quality of life. Studies indicate that individuals with OSA and EDS face up to a 5.6 times higher risk of absenteeism in the USA and a 1.4 times higher risk in Norway, with 23% experiencing long-term occupational adjustments like job changes or altered schedules [35]. A study of 10,330 people over an average of 8.3 years found that EDS independently increases the risk of cardiovascular death by 2.5 to 2.9 times [36]. Subjective measures, such as questionnaires, include the Epworth Sleepiness Scale (ESS), Stanford Sleepiness Scale (SSS), and Karolinska Sleepiness Scale (KSS). The ESS is a validated, reliable, and widely used tool for assessing excessive daytime sleepiness. Validation shows it effectively distinguishes between healthy individuals and those with sleep disorders like OSA. Although its correlation with objective tests, like the MSLT, is modest, the ESS remains the most practical and validated subjective assessment for both clinical and research purposes [3].

In our cohort, baseline EDS was present in 68.9% of participants (ESS ≥ 10), a rate higher than the 56% moderate sleepiness rate reported in recent ESADA registry data (5), possibly reflecting referral bias or regional phenotypic differences. Following six months of CPAP therapy, mean ESS scores decreased substantially from 14 to 8.6 ± 3.7, with residual EDS.

A meta-analysis of randomized controlled studies involving 7332 patients found that CPAP therapy significantly reduces objective excessive daytime sleepiness and enhances measures of alertness, depending on the severity of the condition. Improvements were observed in individuals under 50 with a BMI of 30 kg/m^2^ or higher, a baseline ESS of 11 or more points, and nightly therapy for more than two months, especially in mild OSA patients. In cases of severe OSA, CPAP markedly reduces EDS, with greater improvements noted in ESS scores among patients with higher baseline BMIs, ESS scores ≥ 11, and adherence to treatment of at least three hours per night. Objective vigilance measures also show improvement, with the Multiple Sleep Latency Test (MSLT) increasing by 1.23 min and the Maintenance of Wakefulness Test (MWT) by 1.6 min; these effects are especially prominent in younger patients (<50 years) and those with a BMI ≥ 33 kg/m^2^ [36]. Within the ESADA cohort, residual EDS (ESS > 10) persisted in approximately 28% of patients at the initial follow-up despite CPAP therapy. The highest prevalence (40%) was observed within the first three months, decreasing to between 13% and 19% after four to twenty-four months [5]. Our data confirmed that higher CPAP compliance significantly predicted greater ESS reduction, reinforcing adherence as a primary therapeutic target rather than a secondary concern.

Treatment adherence was generally high in our study population, with a mean CPAP compliance of 87.7% and a median of 95%. Higher compliance was significantly associated with a greater reduction in ESS (*p* = 0.027) and showed a positive trend with improvements in PHQ-9 scores, while residual AHI showed no correlation with success across any PRO.

Our findings have several practical implications for clinical practice: routine multidomain screening, realistic patient counseling, structured reassessment protocols, and adherence as a key target in parallel with efforts to reduce residual AHI.

## 5. Conclusions

Beyond objective physiological parameters, OSA impairs multiple dimensions of daily functioning and societal participation. The multidomain burden documented in our cohort (excessive daytime sleepiness in 68.9%, cognitive impairment in 98.6% and mood disturbances in over 60%) translates directly into reduced occupational productivity, increased workplace absenteeism, high risk of work-related accidents, and withdrawal from social engagement. These functional impairments create a bidirectional relationship: OSA-related symptoms compromise professional and personal functioning, while unemployment and social isolation exacerbate metabolic and psychiatric comorbidities, perpetuating a vicious cycle of disability.

This single-center cohort study of adults with moderate to severe OSA revealed a significant multidomain burden at the time of diagnosis and demonstrated substantial, though incomplete, improvement after six months of CPAP therapy. Notably, one of the most affected aspects was excessive daytime sleepiness, which decreased after six months, with the ESS score dropping to 8.6 ± 3.7; however, residual excessive daytime sleepiness was still present in 27.0% of the participants. The high prevalence of cognitive impairment, assessed via the MoCA-Blind questionnaire, improved to 25.3 ± 2.7; nevertheless, it affected 48.6% of the study population. The rates of mood disturbances, evaluated through the PHQ-9 and GAD-7 scales, decreased to 28.4% and 22.9%, respectively. Quality of life also improved, with the main domains of the WHOQOL-BREF increasing at six months, although a few patients scored ≤50. Higher adherence to CPAP therapy was associated with a greater reduction in ESS scores and tended to lower PHQ-9 scores, reinforcing adherence as a key modifiable factor influencing patient-centered outcomes.

From a clinical perspective, these findings reinforce adherence as one of the top targets in CPAP management and necessitate routine multidomain screening at diagnosis, patient counseling regarding expected benefits and residual symptoms, and structured reassessment protocols monitoring PRO, adherence, and residual respiratory indices. From a public health perspective, optimizing CPAP adherence through targeted interventions may reduce healthcare utilization, restore workforce participation, and alleviate the substantial societal burden of untreated sleep-disordered breathing.

## 6. Study Limitation

Current literature often presents results from patient reassessment at 3 months and, subsequently, at 6 months, comparing scores with those at the time of diagnosis. Considering the incomplete evaluation of some patients in the selected cohort, our study presents results at 6 months following the introduction of CPAP therapy.

Another limitation could stem from the subjectivity in completing questionnaires. Most questions focus on symptoms that are difficult to quantify. Also, the patient’s education level impacts their understanding of the question and, consequently, the formulation of their response.

Our retrospective, single-center study did not include standardized anthropometric data, such as BMI and neck circumference, which limits phenotypic characterization and the ability to compare our findings with those of other studies. Additionally, comorbidity profiling using the Carlson comorbidity index was not systematically available due to the study’s design. Consequently, we chose not to analyze sporadically available data entries to avoid bias. It is important to emphasize that the lack of this data does not affect the study’s primary objectives, namely, the within-patient changes in multidomain patient-reported outcomes after CPAP and their relationships with adherence and residual AHI. We are currently implementing standardized comorbidity collection in ongoing prospective extensions.

Given the growing demand, the department responsible for managing sleep-related respiratory disorders at our hospital is also currently evaluating pediatric patients. Unfortunately, for the moment, only a limited number of cases have been thoroughly investigated both at the time of diagnosis and at 6 months following the initiation of CPAP therapy.

## 7. Further Directions

The purpose of the paper is to identify the extent to which nocturnal respiratory events and their associated symptoms impact cognitive function, psycho-emotional status, and quality of life. At the same time, the research examines how specific therapy can restore standard architecture.

The study remains open and will later include more patients who will be evaluated at the time of sleep apnea diagnosis, as well as at 3 and 6 months after the introduction of CPAP. Furthermore, the aim is to assess pediatric patients in the same manner.

The beneficial effects of CPAP therapy on the evaluated symptoms and on quality of life may manifest over different time intervals. Patients will be reassessed at each CPAP follow-up visit, and results will be periodically documented. Longer follow-up periods are needed to determine the durability of these improvements.

As future research directions, prospective studies with extended follow-up are needed to determine whether cognitive gains plateau after six months or continue to progress over longer periods, ideally supported by longitudinal neuroimaging to map structural recovery and identify predictive biomarkers. Incorporating health economics and implementation science perspectives through cost-effectiveness analyses and evaluation of real-world.

## Figures and Tables

**Figure 1 biomedicines-13-02628-f001:**
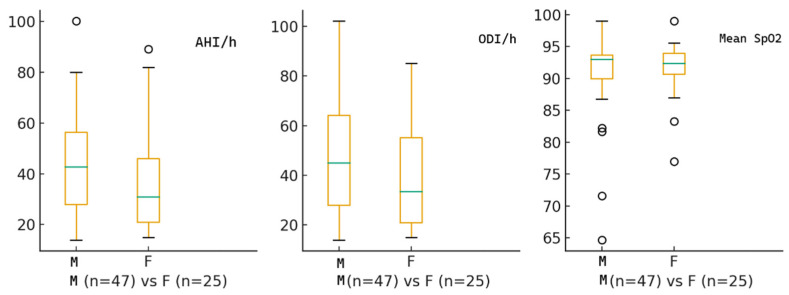
Illustration of the distribution of baseline polygraphy parameters, Apnea–Hypopnea Index (AHI), Oxygen Desaturation Index (ODI), and Mean nocturnal SpO_2_—stratified by sex (M = males, F = females).

**Figure 2 biomedicines-13-02628-f002:**
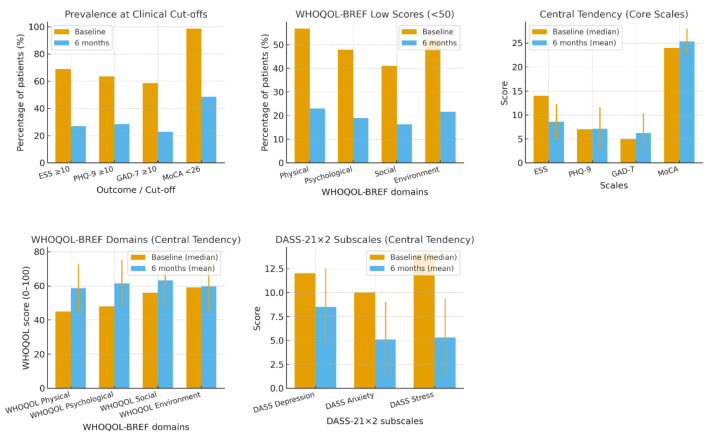
Univariate patient-reported outcomes before and after CPAP (*n* = 72). Prevalence at clinical cut-offs (percentage of patients meeting thresholds): ESS ≥ 10 (EDS), PHQ-9 ≥ 10 (moderate+ depression), GAD-7 ≥ 10 (clinically significant anxiety), MoCA < 26 (cognitive impairment). WHOQOL-BREF low scores (<50) for Physical, Psychological, Social, and Environmental domains. Central tendency of continuous scales (ESS, PHQ-9, GAD-7, MoCA; WHOQOL-BREF domains; DASS-21 × 2 subscales). Bars show baseline medians (orange) and 6-month means ± SD (blue); error bars in C–E depict SD at 6 months. A positive change indicates improvement (lower scores are better for ESS/PHQ-9/GAD-7/DASS; higher scores are better for MoCA and WHOQOL) domains.

**Table 1 biomedicines-13-02628-t001:** Summary of the inclusion and exclusion criteria for participants included in the analysis.

Category	Criterion	Definition/Notes
Inclusion	Age	≥18 years
Inclusion	Confirmed sleep apnea	Diagnosis with AHI ≥ 5 events/h
Inclusion	CPAP adherence	Documented adherence to CPAP therapy (per study criteria; device-reported)
Inclusion	No prior OSA/CPAP	No previous diagnosis of OSA and no prior CPAP treatment
Exclusion	Inability to consent	Unable to sign informed consent or understand the study purpose
Exclusion	Neurological or psychiatric disorders	Any known neurological or psychiatric condition
Exclusion	Acute intercurrent illness	Any acute condition present at enrollment
Exclusion	Incomplete questionnaires	Missing data in required instruments (questionnaires)

**Table 2 biomedicines-13-02628-t002:** Baseline demographic, clinical, psychological, cognitive, and quality-of-life characteristics of the study population (*n* = 72), stratified by sex.

Domain	Metric	Overall (Baseline)	Male (*n* = 47)	Female (*n* = 25)	*p*-Value (Male vs. Female)
Demographics	Age, years—median; mean ± SD	57; 57.1 ± 12.0	52.12 ± 12.44	61.08 ± 10.00	0.009
	Sex, n (%)	Male 47 (65.3%); Female 25 (34.7%)	48 (100%)	25 (100%)	—
Respiratory indices	AHI, events·h^−1^—median	34.5	43.65	31.00	0.263
	ODI, events·h^−1^—median	35.5	46.05	33.50	0.208
	Mean nocturnal SpO_2_, %—median	92.4	92.70	92.40	0.764
Sleepiness	ESS, 0–24—median	14	12.88 ± 4.59	13.60 ± 4.95	0.597
Mood/anxiety	PHQ-9, 0–27—median	7	8.50	13.00	0.074
	GAD-7, 0–21—median	5	6.50	12.00	0.011
Cognition	MoCA, 0–30—median	24	18.50	17.00	0.031
DASS-21 (×2)	Depression—median	12	7.00	12.00	0.039
	Anxiety—median	10	—	—	—
	Stress—median	14	11.00	12.00	0.294
WHOQOL-BREF (0–100)	Physical—median	45	44.00	45.00	0.732
	Psychological—median	48	56.00	45.00	0.779
	Social—median	56	60.00	56.00	0.936
	Environment—median	59	56.96 ± 13.01	56.48 ± 13.48	0.900
Baseline prevalence (per preset cut-offs)	EDS (ESS ≥ 10), %	68.9%	79.2%	76.0%	0.772
	PHQ-9 ≥ 10 (moderate + depression), %	63.5%	45.8%	72.0%	0.441
	GAD-7 ≥ 10 (clinically significant anxiety), %	58.6%	37.5%	68.0%	0.221
	MoCA < 26 (impairment), %	98.6%	100.0%	100.0%	1.000
	WHOQOL-BREF < 50 (low), Physical/Psychological/Social/Environment, %	56.8/47.9/41.1/52.1	62.5/33.3/12.5/29.2%	64.0/56.0/16.0/28.0%	0.464/0.145/1.000/0.564

**Table 3 biomedicines-13-02628-t003:** Summary of the evolution of symptom severity, cognitive performance, and quality-of-life scores from baseline (T1) to the 6-month follow-up (T2), stratified by sex.

Metric	Male T1	Female T1	p T1	Male T2	Female T2	p T2	n (M/F) 47/25
ESS	13.23 ± 4.60	13.60 ± 4.95	0.757	6.00	8.00	0.730	47/25
PHQ-9	12.00	13.00	0.325	7.00	7.00	0.317	47/25
GAD-7	10.00	12.00	0.116	4.00	5.00	0.204	47/25
MoCA	18.00	17.00	0.056	21.00	21.00	0.397	47/25
WHO Physical	48.00	45.00	0.313	79.00	64.00	0.041	47/25
WHO Psychological	53.71 ± 15.95	47.60 ± 16.89	0.142	73.50	70.00	0.229	47/25
WHO Social	56.00	56.00	0.944	70.00	67.00	0.052	47/25
WHO Environment	59.98 ± 13.71	56.48 ± 13.48	0.300	79.44 ± 12.20	67.84 ± 11.73	0.000	47/25

## Data Availability

Data unavailable due to privacy or ethical reasons.

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
