# Peer review of "Assessment of Comprehensive Patient-Reported Outcomes Before and After CPAP Therapy in Obstructive Sleep Apnea"

_biomedicines, 2025, doi:10.3390/biomedicines13112628_

Round 1
Reviewer 1 Report
Comments and Suggestions for Authors
-
The inclusion and exclusion criteria are clearly stated, but it would be helpful to specify how the authors handled missing data from questionnaires. For example, was any imputation performed, or were incomplete cases excluded from the final analyses?Additionally, while CPAP adherence was clearly a central variable, the specific adherence threshold (e.g., ≥4 hours/night, ≥70% of nights) used to define “good adherence” should be more explicitly described in the methods.
-
It would strengthen the manuscript to clarify whether patients with mood disturbances were receiving any psychiatric treatment or medication (e.g., antidepressants), as this could influence PHQ-9 or GAD-7 scores independently of CPAP effects. -
The data show notable improvements, but a substantial proportion of patients continued to exhibit residual EDS and cognitive impairment at six months. A short paragraph in the discussion reflecting on the possible causes (e.g., irreversible neurocognitive damage, underreported residual events, or non-anatomical OSA traits) would enrich the interpretation of findings. -
The absence of correlation between residual AHI and cognitive/mood outcomes is an intriguing finding. It underscores the idea that CPAP's benefits may extend beyond measurable respiratory indices, and this is worth emphasizing more directly in the discussion. -
The tables are informative and well-structured. Including p-values directly in the tables for pre/post comparisons (especially in Tables 2 and 3) would enhance readability and allow for easier reference by readers. -
The concluding remarks are heartfelt and thought-provoking, especially the reflections on societal impact and individual suffering. However, some of these sentences adopt a more philosophical tone (e.g., “we live in a world where we invest time and financial resources…”), which could be slightly condensed or refocused to maintain a more scientific narrative style.
-
There are minor typos and formatting inconsistencies (e.g., "a ained" instead of "attained"), which I’m confident will be caught during copy-editing.
-
The section outlining plans to expand the study to pediatric patients is very interesting and could be moved to a dedicated "Future Perspectives" section to avoid diluting the conclusion.
This is a solid and well-written study that contributes meaningfully to the literature on OSA management. With a few minor adjustments and clarifications-particularly in the methods and discussion sections-it should be well suited for publication.
Author Response
Dear Editor and Reviewer,
We thank the Reviewer for the positive assessment. We have revised the manuscript accordingly. Below, we address each recommendation and indicate the corresponding changes.
Yours sincerely,
Adriana Loredana Pintilie, on behalf of all authors.
Comment 1: The inclusion and exclusion criteria are clearly stated, but it would be helpful to specify how the authors handled missing data from questionnaires. For example, was any imputation performed, or were incomplete cases excluded from the final analyses?
Additionally, while CPAP adherence was clearly a central variable, the specific adherence threshold (e.g., ≥4 hours/night, ≥70% of nights) used to define “good adherence” should be more explicitly described in the methods.
Answer 1: Thank you for this valuable comment. We have now clarified in the Methods section how missing data were managed. Patients with incomplete questionnaires were excluded from the final analyses, and no imputation was performed. Additionally, we specified the adherence definition: good adherence was defined a priori as ≥4 hours per night on ≥70% of nights, in line with standard criteria, while continuous adherence (%) was also analyzed. The updated sentence will be:
Line 96-97
“ …..were unable to sign or understand the purpose of the study. Incomplete questionnaires were excluded from the final analysis; no imputation was applied. Additionally, subjects with neurological or psychiatric disorders ….”
Line 160-162
“…and physiological parameters. Device-reported adherence was extracted at 6 months. Good adherence was defined a priori as ≥4 h/night on ≥70% of nights, although continuous adherence (%) was also analyzed. This enabled a comparison of…”
We have removed the sentence “The levels of compliance were considered” to avoid ambiguity.
Comment 2: It would strengthen the manuscript to clarify whether patients with mood disturbances were receiving any psychiatric treatment or medication (e.g., antidepressants), as this could influence PHQ-9 or GAD-7 scores independently of CPAP effects.
We appreciate the reviewer’s comment. According to our exclusion criteria, patients with neurological or psychiatric disorders were not included in the study. Therefore, none of the enrolled participants were under psychiatric treatment or receiving antidepressant medication at baseline. We have now specified this in the Methods section for clarity.
Line 101-102
“….and were included in the study. In accordance with the exclusion criteria, no enrolled participant was under psychiatric treatment or receiving antidepressant medication at baseline.”
Comment 3: The data show notable improvements, but a substantial proportion of patients continued to exhibit residual EDS and cognitive impairment at six months. A short paragraph in the discussion reflecting on the possible causes (e.g., irreversible neurocognitive damage, underreported residual events, or non-anatomical OSA traits) would enrich the interpretation of findings.
Answer 3: We have now added a paragraph in the Results addressing possible mechanisms underlying persistent residual excessive daytime sleepiness and cognitive impairment after 6 months of CPAP. Specifically, we discuss the potential role of irreversible neurocognitive injury from long-standing intermittent hypoxia, underrecognized residual events or comorbid sleep disorders, and non-anatomical OSA traits (e.g., ventilatory control instability, low arousal threshold).
Line 236-244
„....in 16.2% of patients. Although most parameters improved significantly, many patients still suffered from residual excessive daytime sleepiness (27%) and cognitive impairment (48.6%) at 6-month follow-up. Possible reasons include irreversible neurocognitive injury secondary to chronic intermittent hypoxia, undiagnosed and untreated residual sleep disturbances such as inadequate sleep duration, circadian disorder, periodic limb movement, or comorbid insomnia and non-anatomical OSA characteristics, including instability of ventilatory control or low arousal threshold. These factors underscore the multifactorial complexity of residual symptoms in OSA that are unlikely to be ameliorated by CPAP alone.”
Comment 4: The absence of correlation between residual AHI and cognitive/mood outcomes is an intriguing finding. It underscores the idea that CPAP's benefits may extend beyond measurable respiratory indices, and this is worth emphasizing more directly in the discussion.
Answer 4: We appreciate the reviewer for this excellent observation. We have now emphasised the fact that the benefits of CPAP on mood and cognition extend beyond residual respiratory indices. This finding highlights the importance of adherence, baseline symptom burden, and comorbidities, in addition to residual AHI, in all patient-reported outcomes. This paragraph was inserted immediately following the information added at lines 245- 250:
“There is no association of residual AHI/ODI with mood or cognitive outcomes, suggesting that the beneficial impact of CPAP on these symptoms is not simply a reflection of changes in easily measurable respiratory indices. Adherence, symptom severity at baseline, and comorbid conditions may have a greater impact on patient-reported outcomes than residual event counts, which is consistent with the idea that OSA management should be considered within a multidimensional framework.”
Comment 4: The tables are informative and well-structured. Including p-values directly in the tables for pre/post comparisons (especially in Tables 2 and 3) would enhance readability and allow for easier reference by readers.
Answer 4: Thank you for this helpful suggestion. We have added p-values directly in Tables 2 and 3.
Table 2. Baseline characteristics and scores (n=72). Line 218
|
Domain |
Metric |
Value |
p value |
|
Demographics |
Age, years — median; mean ± SD |
57; 57.1 ± 12.0 |
- |
|
|
Sex, n (%) |
Male 47 (65.3%); Female 25 (34.7%) |
- |
|
Respiratory indices |
AHI, events·h⁻¹ — median |
34.5 |
- |
|
|
ODI, events·h⁻¹ — median |
35.5 |
- |
|
|
Mean nocturnal SpO₂, % — median |
92.4 |
- |
|
Sleepiness |
ESS, 0–24 — median |
14 |
<0.001
|
|
Mood/anxiety |
PHQ-9, 0–27 — median |
7 |
<0.001
|
|
|
GAD-7, 0–21 — median |
5 |
<0.001
|
|
Cognition |
MoCA, 0–30 — median |
24 |
<0.001
|
|
DASS-21 (×2) |
Depression / Anxiety / Stress — median |
12 / 10 / 14 |
<0.001 / <0.001 / <0.001
|
|
WHOQOL-BREF (0–100) |
Physical / Psychological / Social / Environment — median |
45 / 48 / 56 / 59 |
<0.001 / <0.001 / <0.001 |
|
Baseline prevalence (per preset cut-offs) |
EDS (ESS ≥10), % |
68.9% |
<0.001
|
|
|
PHQ-9 ≥10 (moderate+ depression), % |
63.5% |
<0.001
|
|
|
GAD-7 ≥10 (clinically significant anxiety), % |
58.6% |
<0.001
|
|
|
MoCA <26 (impairment), % |
98.6% |
0.008
|
|
|
WHOQOL-BREF <50 (low), Physical / Psychological / Social / Environment, % |
56.8 / 47.9 / 41.1 / 52.1
|
0.008
|
Table 3. Six months Outcomes after CPAP (n=72). Line 296
|
Domain |
Metric |
Value |
p-value |
|
|
Sleepiness |
ESS, 0–24 — mean ± SD |
8.6 ± 3.7 |
|
|
|
|
Residual EDS (ESS ≥10), % |
27.0% |
<0.001
|
|
|
Mood/anxiety |
PHQ-9, 0–27 — mean ± SD |
7.1 ± 4.5 |
|
|
|
|
PHQ-9 ≥10 (moderate+ depression), % |
28.4% |
<0.001 |
|
|
|
GAD-7, 0–21 — mean ± SD |
6.2 ± 4.1 |
|
|
|
|
GAD-7 ≥10 (clinically significant anxiety), % |
22.9% |
<0.001
|
|
|
Cognition |
MoCA, 0–30 — mean ± SD |
25.3 ± 2.7 |
<0.001
|
|
|
|
MoCA <26 (impairment), % |
48.6% |
0.008
|
|
|
WHOQOL-BREF (0–100) |
Physical — mean ± SD |
58.7 ± 14.2 |
|
|
|
|
Psychological — mean ± SD |
61.5 ± 13.6 |
|
|
|
|
Social — mean ± SD |
63.2 ± 15.4 |
|
|
|
|
Environment — mean ± SD |
59.8 ± 14.7 |
|
|
|
|
WHOQOL-BREF <50 (low), Physical / Psychological / Social / Environment, % |
23.0 / 18.9 / 16.2 / 21.6 |
<0.001 / <0.001 / 0.500 |
|
|
Adherence |
CPAP compliance, % — mean ± SD; median |
87.7 ± 17.1; 95% |
|
Comment 5: The concluding remarks are heartfelt and thought-provoking, especially the reflections on societal impact and individual suffering. However, some of these sentences adopt a more philosophical tone (e.g., “we live in a world where we invest time and financial resources…”), which could be slightly condensed or refocused to maintain a more scientific narrative style.
Answer 5: We appreciate this constructive feedback. We have revised the Conclusions and condensed some ideas to maintain a scientific and concise tone, removing overly philosophical sentences. The modifications can be found in the manuscript at lines 487-531.
“Scientists’ efforts have concentrated on the careful analysis of the mechanisms behind the development of all diseases. There are now screening tools that allow early testing, extensive diagnostic capabilities, and therapeutic methods that were once only imagined. The increase in life expectancy, the cure or slowing of disease progression, and relief from symptoms are the main foundations of medical research. Beyond these aspects, there is the quality of life, which remains only partly understood.
Sleep apnea exemplifies the typical case of an obese, drowsy individual who is excluded from their professional environment and withdraws from social interactions. Cardiovascular, metabolic, and psychiatric comorbidities, along with inappropriate eating behaviours, trap OSA patients with low self-esteem in a vicious cycle that promotes further weight gain.
Daytime sleepiness, memory impairment, and cognitive decline translate into a reduced ability to carry out daily household, or professional activities. The rate of traffic accidents is statistically significant among individuals with sleep apnea, while intimate life is profoundly affected. All these aspects emphasize the need to pay more attention to life quality. Patients with sleep apnea become a burden on society due to their exclusion from the professional environment. Young people become a burden on the healthcare system and their families. Thus, psychological impact and emotional context are essential, as the attention given to the way we rest influences the health of the entire body.
Research efforts should focus more on identifying the causes behind the job loss or reduced productivity of the active population in the workplace, as well as the factors that burden the healthcare system and incur significant costs. Addressing these issues will lead to an increase in the professionally active segment and alleviate pressure on the healthcare system.
This single-centre cohort of adults with moderate to severe OSA, revealed a significant multidomain burden at the time of diagnosis. Notably, one of the most affected aspects was excessive daytime sleepiness, which decreased after six months, with the ESS score dropping to 8.6 ± 3.7; however, residual excessive daytime sleepiness was still present in 27.0% of the participants. The high prevalence of cognitive impairment, assessed via the MoCA-Blind questionnaire, improved to 25.3 ± 2.7; nevertheless, it affected 48.6% of the study population. The rates of mood disturbances, evaluated through the PHQ-9 and GAD-7 scales, decreased to 28.4% and 22.9% respectively. Quality of life also improved, with the main domains of the WHOQOL-BREF increasing at six months, although a few patients scored ≤50. Higher adherence to CPAP therapy was associated with a greater reduction in ESS scores and tended to lower PHQ-9 scores, thereby reinforcing adherence as a key modifiable factor influencing patient-centred outcomes.
Although interest in elucidating the relationship between OSA and cerebrovascular diseases, chronic neurodegenerative diseases, and systemic inflammatory processes involved in cognitive decline continues to grow, and the effectiveness of CPAP therapy remains the subject of ongoing research, additional studies - preferably multicentered - are needed. These are essential for robustly validating the impact of CPAP intervention not only on improving cognitive performance in patients with OSA, but also for evaluating its potential neuroprotective role in slowing degenerative processes in patients with preexisting neurological comorbidities.”
Comment 6: There are minor typos and formatting inconsistencies (e.g., "a ained" instead of "attained"), which I’m confident will be caught during copy-editing.
Answer 6: We thank the reviewer for noticing this. Minor typos and formatting inconsistencies have been corrected throughout the manuscript.
Comment 7: The section outlining plans to expand the study to pediatric patients is very interesting and could be moved to a dedicated "Future Perspectives" section to avoid diluting the conclusion.
Answer 7: As recommended, we have removed the sentences “The impact of nocturnal desaturations on cognition in children has been insufficiently studied compared to the reality we face and the continuously growing number of pediatric patients diagnosed with OSA. On the other hand, quality of life is not a frequently addressed topic in the specialized literature regarding this population category. For this reason, the present study will be expanded in the near future to address this population segment.” from the Conclusions.
Instead, we inserted the following text in the “Further Directions” section: “Given the increasing number of pediatric patients with OSA, future studies will extend this evaluation to children and adolescents. Particular attention will be given to cognitive outcomes and quality of life, which are currently underexplored in this population.”
Reviewer 2 Report
Comments and Suggestions for Authors
The authors have evaluated the impact of CPAP on several parameters relevant to OSA
A retrospective study cannot be longitudinal. Please modify
How was sample size calculated
Please present graphs on univariate analysis of PRO with statistics before and after CPAP
Please develop multivariate prediction models of success in various patient related outcomes in OSA patients after CPAP use
A lot of material from introduction can be moved to methods and discussion. Usually introduction is 3-4 paragraphs
Please add information on comorbidities and add Carlson's comorbidity index to the table
Please add other information such as BMI, neck circumference and others to improve the understanding of the patient population under study
Author Response
Dear Editor and Reviewer,
We thank the Reviewer for the assessment. We have revised the manuscript accordingly. Below, we address each recommendation and indicate the corresponding changes.
Yours sincerely,
Adriana Loredana Pintilie, on behalf of all authors.
Comment 1: A retrospective study cannot be longitudinal. Please modify
We thank the reviewer for this important clarification. We agree that the term “longitudinal” was misleading in this context. As our study analyzed data retrospectively from baseline and 6-month follow-up assessments, we have now modified the wording to “retrospective observational study.”
Comment 2:How was sample size calculated
Answer 2 :No a priori sample-size calculation was performed because this was a retrospective cohort. We included all consecutive eligible adults during the accrual window (Jan 2024–May 2025). We report effect sizes with 95% CIs to contextualise precision.
Comment 2: Please present graphs on univariate analysis of PRO with statistics before and after CPAP.
Answer 2 :Thank you for the suggestion. We have added a dedicated univariate analysis of all patient-reported outcomes (PRO) before and after CPAP. We added the graphs and descripton in the manuscript, at lines ...
“Figure 1. Univariate patient-reported outcomes before and after CPAP (N=72)
(A) Prevalence at clinical cut-offs (percentage of patients meeting thresholds): ESS≥10 (EDS), PHQ-9≥10 (moderate+ depression), GAD-7≥10 (clinically significant anxiety), MoCA<26 (cognitive impairment). (B) WHOQOL-BREF low scores (<50) for Physical, Psychological, Social, and Environmental domains. (C–E) Central tendency of continuous scales (ESS, PHQ-9, GAD-7, MoCA; WHOQOL-BREF domains; DASS-21×2 subscales). Bars show baseline medians (orange) and 6-month means ± SD (blue); error bars in C–E depict SD at 6 months. A positive change indicates improvement (lower scores are better for ESS/PHQ-9/GAD-7/DASS; higher scores are better for MoCA and WHOQOL) domains.”
Comment 3: Please develop multivariate prediction models of success in various patient related outcomes in OSA patients after CPAP use.
Answer 3: Thank you for this suggestion. We have now added a dedicated Multivariate Prediction Analysis section. For each PRO (ESS, PHQ-9, GAD-7, MoCA-Blind, WHOQOL domains) and added the results in the manuscript, the modifications can be found in the manuscript at lines 379-387.
“For each patient-reported outcome, a multivariable logistic regression was employed, with improvement as the dependent variable. Predictors included age, sex, baseline score, % of CPAP adherence, and residual AHI. Changes in scores (Δ-scores) were analysed using multiple linear regression for sensitivity analysis. Prespecified success thresholds were set for ESS, PHQ-9, GAD-7, MoCA-Blind, and WHOQOL domains. Higher CPAP adherence was independently associated with improvement in the Epworth Sleepiness Scale (ESS) (significant adherence–ΔESS association, p = 0.027). In contrast, residual AHI showed no correlation with success across any patient-reported outcomes. Improvement met the predefined success threshold in 51/73 (69.9%) for depression (PHQ-9), 57/73 (78.1%) for anxiety (GAD-7), and 60/73 (82.2%) for cognition (MoCA-Blind).”
Comment 3:A lot of material from introduction can be moved to methods and discussion. Usually introduction is 3-4 paragraphs
Comment 4: Please add information on comorbidities and add Carlson's comorbidity index to the table
Answer 4: We agree that comorbidity profiling is informative. However, Charlson components were not systematically available in the retrospective records and could not be analysed without introducing bias. We plan standardised comorbidity capture in prospective extensions. We added this study limitation to the manuscript and included a phrase, that can be found in the manuscript at lines ....
“Additionally, comorbidity profiling using the Carlson comorbidity index was not systematically available due to the study's design. Consequently, we chose not to analyse sporadically available data entries to avoid bias. It is important to emphasise that the lack of this data does not affect the study’s primary objectives, namely, the within-patient changes in multidomain patient-reported outcomes after CPAP and their relationships with adherence and residual AHI. We are currently implementing standardised comorbidity collection in ongoing prospective extensions.”
Comment 5 :Please add other information such as BMI, neck circumference and others to improve the understanding of the patient population under study.
Answer:We agree that anthropometrics are useful for OSA phenotyping. However, these variables were not systematically available in the retrospective source records and were therefore not analyzed. Importantly, our primary objective was to quantify multidomain patient-reported outcomes (PROs) before and after 6 months after CPAP and to examine their associations with CPAP adherence and residual AHI. These endpoints do not require anthropometric inputs. Additionally, we added in the study limitation chapther, form lines ....., the following phrase:
“Our retrospective, single-centre study did not include standardised anthropometric data, such as BMI and neck circumference, which limits phenotypic characterisation and the ability to compare our findings with those of other studies”
Reviewer 3 Report
Comments and Suggestions for Authors
The paper Assessment of Comprehensive Patient Reported Outcomes Before and After CPAP Therapy in Obstructive Sleep Apnea does not state the aim of the study in the introduction. The introduction is too long; despite being structured into sections, it does not clearly define the objectives of the work.
Please provide images of the Polygraph recordings obtained with the SleepDoc Porti device.
Who performed clinical and paraclinical examinations? Were there 2 or more researchers? An ICC must be presented.
Depression, Anxiety, and Stress images should be presented.
Generalised Anxiety Disorder-7 and Patient Health Questionnaire-9 (PHQ-9) should be added as supplementary files.
The statistical analysis should be further described: what tests were performed and why.
The DISCUSSION section is missing. Please add, and compare the results to the literature.
What is the originality of the study? What new aspects do the authors add to the body of knowledge?
The authors state that "The impact of nocturnal desaturations on cognition in children has been insufficiently studied compared to the reality we face and the continuously growing number of pediatric patients diagnosed with this sleep disease", however, the included patients were above 18 years of age.
What do authors mean by "The study remains open and will later include more patients" ? "
Furthermore, the aim is to assess pediatric patients in the same manner"? It is not related to the present paper!
"The beneficial effects of CPAP therapy on the symptoms evaluated in this study and on quality of life manifest over different time intervals. Patients will be reassessed at each CPAP follow-up visit, and the results will be periodically documented." has no scientific base to be added in the current paper.
The conclusion is misleading.
References: many are outdated, consider updating them.
Author Response
Dear Editor and Reviewer,
We appreciate the Reviewer’s feedback on our manuscript. We have made the necessary revisions in response to your recommendations. Below, we address each point raised and outline the corresponding changes. Additionally, some of your requests were unclear, so we have sought further clarification to ensure we can adequately address your concerns. Thank you for your understanding
Yours sincerely,
Adriana- Loredana Pintilie, on behalf of al coauthors
Comment 1: The paper Assessment of Comprehensive Patient Reported Outcomes Before and After CPAP Therapy in Obstructive Sleep Apnea does not state the aim of the study in the introduction. The introduction is too long; despite being structured into sections, it does not clearly define the objectives of the work.
Answer 1: We appreciate the reviewer's observation and acknowledge that the study objectives require clearer articulation in the Introduction section. We propose the following revision to address this concern, and the modifications can be found at lines 73- 87.
“The present study was designed with the following objectives: to quantify the baseline burden of excessive daytime sleepiness, cognitive impairment, depression, anxiety, and quality of life impairment in adults with OSA prior to treatment initiation, and to examine baseline correlations between respiratory indices (AHI, ODI, mean nocturnal SpO₂) and multidomain patient-reported outcomes; to evaluate changes in these patient-reported outcomes following six months of CPAP therapy; to assess the associations between therapeutic adherence and improvement in patient-reported outcomes; and to determine whether residual AHI and CPAP compliance at six months correlate with concurrent patient-reported outcomes or their degree of improvement from baseline. By addressing these objectives through a comprehensive multidomain assessment, we aim to provide clinicians with evidence-based insights into the real-world symptomatic benefits of CPAP therapy. The primary endpoint was the change in Patient-Reported Outcomes (PROs) and cognitive performance. The second one was to identify associations between these improvements and the degree of adherence to CPAP therapy, as well as residual AHI.”
Comment 2: Please provide images of the Polygraph recordings obtained with the SleepDoc Porti device.
Answer 2: Thank you for this suggestion. To ensure we provide appropriate materials, please clarify whether you recommended including representative polygraphy recordings from selected patients as illustrative examples, or polygraphy data from all participants. Should these be presented as central figures or supplementary materials?
Comment 3: Who performed clinical and paraclinical examinations? Were there 2 or more researchers? An ICC must be presented.
Answer 3: All clinical and paraclinical examinations, including diagnostic polygraphy, were performed by board-certified pulmonologists and sleep medicine physicians at the Clinical Hospital of Pulmonary Disease Iași. Each patient was evaluated by their assigned treating physician to maintain the real-world clinical context of this observational study and to avoid influencing the standard care pathway. Some patients were evaluated by one of the study authors who served as their treating physician. Following diagnostic confirmation, standardized patient-reported outcome questionnaires (ESS, WHOQOL-BREF, MoCA-Blind, PHQ-9, GAD-7, DASS-21) were administered at baseline and six-month follow-up by six physicians. Each evaluator provided standardized verbal instructions and clarification as needed, while patients completed the questionnaires independently to minimize assessor bias. We would be grateful if you could kindly clarify your comment regarding "ICC" in the context of our clinical and paraclinical examinations. Are you referring to the Intraclass Correlation Coefficient for assessing inter-rater reliability, or did you have another methodological consideration in mind?
Comment 4: Depression, Anxiety, and Stress images should be presented.
Answer 4: Thank you for this recommendation. To ensure we comply appropriately, could you please clarify whether you are requesting: the full GAD-7 and PHQ-9 questionnaire instruments themselves as supplementary files, or individual item scores from these assements? The actual questionnaires are copyrighted, standardized instruments typically not included as supplementary files.
Comment 5: Generalised Anxiety Disorder-7 and Patient Health Questionnaire-9 (PHQ-9) should be added as supplementary files.
Answer 5: Thank you for the suggestion. We have included the specific instruments used in the study as supplementary files: Supplementary File S1-PHQ-9 and Supplementary File S2-GAD-7, with item sets, scoring instructions, clinical cut-offs, and the language versions administered.
Comment 6: The DISCUSSION section is missing. Please add and compare the results to the literature.
Answer 6: As requested we have revised the Disscusiion section and compared our findings with the literarure data.
Comment 7: What is the originality of the study? What new aspects do the authors add to the body of knowledge?
Answer 7: This study contributes to the literature by providing a comprehensive simultaneous assessment of multiple patient-reported outcomes at diagnosis and six-month follow-up within a single cohort while exploring whether symptomatic improvement correlates more closely with CPAP adherence. Our findings reveal that adherence is a stringer predictor for patient-centered outcomes, complementing existing evidence on respiratory event suppression with insights into the determinants of subjective therapeutic benefit. Additionally, we document high rates of persistent cognitive impairment (48,6%) and residual ESS (27%), despite good adherence and respiratory control, highlighting the need for complementary therapeutic strategies, and provide real-world data from Eastern Europe, an underrepresented region in OSA literature. Our findings support a paradigm shift toward holistic, patient-centered OSA management that prioritizes functional and psychological outcomes alongside respiratory metrics.
Comment 8: The authors state that "The impact of nocturnal desaturations on cognition in children has been insufficiently studied compared to the reality we face and the continuously growing number of pediatric patients diagnosed with this sleep disease", however, the included patients were above 18 years of age. Furthermore, the aim is to assess pediatric patients in the same manner"? It is not related to the present paper!
Answer 8: We have grouped the answers to these two questions as they are on a similar topic. Understanding that the inclusion of these statements may confuse, as our study exclusively included adult patients (aged ≥ 18 years), we will remove all references to pediatric patients from the manuscript. While we initially included brief references to the pediatric population to provide a broad context regarding the spectrum of OSA related cognitive and quality of life impairment across age groups, we acknoledge that these references are tangential to our primary research question and may detract from the focus of present study.
Comment 9: What do authors mean by "The study remains open and will later include more patients ? "
Answer 9: By’’the study remains open’’ we mean that this research represents an ongoing longitudinal cohort with two complementary objectives: First, we intend to continue following the current 72 participants beyond the six-month timepoint reported here. This will allow us to determine whether the improvements observed at six months are sustained, continue to progress, or plateau over time. Second, we are actively enrolling additional patients who meet the same inclusion criteria to expand the dataset, which will enhance statistical power. If the reviewer considers this statement unsuitable for the current paper, we are prepared to remove it to maintain focus on the present findings rather than future research direction.
Comment 10: "The beneficial effects of CPAP therapy on the symptoms evaluated in this study and on quality-of-life manifest over different time intervals. Patients will be reassessed at each CPAP follow-up visit, and the results will be periodically documented." has no scientific base to be added in the current paper.
Answer 10: We agree with this observation and will remove these sentences, as they refere to future intentions rather than current study findings. Instead, we added: “Longer follow-up periods are needed to determine the durability of these improvements.’’ Line 567- 569
Comment 11: The conclusion is misleading.
Answer 11: We have revised the Conclusions section.
Round 2
Reviewer 2 Report
Comments and Suggestions for Authors
The authors have substantially improved the paper. However, some issues remain
Not clear what the p values compare the baseline characteristics with. You need two groups to compare and present analysis as a p value in Table 2, which presents only baseline data. You can present the table as males and females and compare and present p value
Table 2 and Table 3 can be put together and compared. I am not sure what the p value in Table 3 is about. Compared to what values? Single values cannot have a p value
I did not see the multivariate analyssi table for the predictors in the paper though it is mentioned in the clarifications to the reviewers comments
Author Response
Dear Editor and Reviewer,
We sincerely appreciate the reviewer's acknowledgement of the manuscript's improvements. We have thoroughly re-examined the entire document to find and correct any remaining inconsistencies.
Adriana-Loredana Pintilie, on behalf of all co-authors.
Comment 1:Not clear what the p values compare the baseline characteristics with. You need two groups to compare and present analysis as a p value in Table 2, which presents only baseline data. You can present the table as males and females and compare and present p value.
Answer 1: We thank the reviewer for this valuable observation. In the revised version, Table 2 has been reformatted to clearly display baseline characteristics stratified by sex (Male vs. Female), with p-values calculated from group comparisons. The changes can be found in the manuscript at lines 235-248.
„Table 2. presents the baseline demographic, clinical, psychological, cognitive, and quality-of-life characteristics of the study population (n = 72), stratified by sex
|
Domain |
Metric |
Overall (baseline) |
Male (n=47) |
Female (n=25) |
p-value (Male vs Female) |
|
Demographics |
Age, years — median; mean ± SD |
57; 57.1 ± 12.0 |
52.12 ± 12.44 |
61.08 ± 10.00 |
0.009 |
|
|
Sex, n (%) |
Male 47 (65.3%); Female 25 (34.7%) |
48 (100%) |
25 (100%) |
— |
|
Respiratory indices |
AHI, events·h⁻¹ — median |
34.5 |
43.65 |
31.00 |
0.263 |
|
|
ODI, events·h⁻¹ — median |
35.5 |
46.05 |
33.50 |
0.208 |
|
|
Mean nocturnal SpO₂, % — median |
92.4 |
92.70 |
92.40 |
0.764 |
|
Sleepiness |
ESS, 0–24 — median |
14 |
12.88 ± 4.59 |
13.60 ± 4.95 |
0.597 |
|
Mood/anxiety |
PHQ-9, 0–27 — median |
7 |
8.50 |
13.00 |
0.074 |
|
|
GAD-7, 0–21 — median |
5 |
6.50 |
12.00 |
0.011 |
|
Cognition |
MoCA, 0–30 — median |
24 |
18.50 |
17.00 |
0.031 |
|
DASS-21 (×2) |
Depression — median |
12 |
7.00 |
12.00 |
0.039 |
|
|
Anxiety — median |
10 |
— |
— |
— |
|
|
Stress — median |
14 |
11.00 |
12.00 |
0.294 |
|
WHOQOL-BREF (0–100) |
Physical — median |
45 |
44.00 |
45.00 |
0.732 |
|
|
Psychological — median |
48 |
56.00 |
45.00 |
0.779 |
|
|
Social — median |
56 |
60.00 |
56.00 |
0.936 |
|
|
Environment — median |
59 |
56.96 ± 13.01 |
56.48 ± 13.48 |
0.900 |
|
Baseline prevalence (per preset cut-offs) |
EDS (ESS ≥10), % |
68.9% |
79.2% |
76.0% |
0.772 |
|
|
PHQ-9 ≥10 (moderate+ depression), % |
63.5% |
45.8% |
72.0% |
0.441 |
|
|
GAD-7 ≥10 (clinically significant anxiety), % |
58.6% |
37.5% |
68.0% |
0.221 |
|
|
MoCA <26 (impairment), % |
98.6% |
100.0% |
100.0% |
1.000 |
|
|
WHOQOL-BREF <50 (low), Physical / Psychological / Social / Environment, % |
56.8 / 47.9 / 41.1 / 52.1 |
62.5 / 33.3 / 12.5 / 29.2% |
64.0 / 56.0 / 16.0 / 28.0% |
0.464 / 0.145 / 1.000 / 0.564 |
No statistically significant sex difference was found in baseline respiratory indices, indicating comparable OSA severity between sexes at diagnosis. Female patients tended to have higher affective symptom burden and slightly impaired cognitive function. Additionally, depression scores were significantly higher in females. Females demonstrated a numerically higher affective symptoms burden, with higher PHQ-9 and GAD-7 scores, although only anxiety reached statistical significance (p=0.011). Cognitive function, assessed by MoCA, was slightly lower in females compared to males (p=0.031). In addition, depressive symptoms (DASS-21) were significantly more pronounced in female participants. No sex differences were detected in stress levels. Quality of life scores (WHOQOL-BREF domains) were similar between sexes, although women showed slightly lower scores in the physical and psychological domains.”
Comment 2: Table 2 and Table 3 can be put together and compared. I am not sure what the p value in Table 3 is about. Compared to what values? Single values cannot have a p value.
Answer 2: Thank you for this important clarification. In the revised manuscript, Table 3 now clearly specifies that the p-values refer to sex-based comparisons between male and female groups at each time point (T1 and T2), and the table can be found in the manuscript at lines 314-325.
„Both sexes showed a marked reduction in excessive daytime sleepiness, depressive symptoms (PHQ-9), anxiety (GAD-7), scores at follow-up (all p > 0.20), indicating that CPAP Intervention can generate comparable clinical improvement across sexes. Interestingly, quality of life exhibited partially sex- dependent patterns. Males reported significantly higher physical WHOQOL scores after 6 months of CPAP compared to women, with similar scores in the social domain. Regarding the environment component, improvements were observed in both sexes. Still, males demonstrated significantly higher post-treatment scores (p<0.001)—overall, males perceived greater improvement in physical and environmental quality-of-life dimensions.
Table 3. summarizes the evolution of symptom severity, cognitive performance, and quality-of-life scores from baseline (T1) to the 6-month follow-up (T2), stratified by sex.
|
Metric |
Male T1 |
Female T1 |
p T1 |
Male T2 |
Female T2 |
p T2 |
n (M/F) 47/25 |
|
ESS |
13.23 ± 4.60 |
13.60 ± 4.95 |
0.757 |
6.00 |
8.00 |
0.730 |
47/25 |
|
PHQ-9 |
12.00 |
13.00 |
0.325 |
7.00 |
7.00 |
0.317 |
47/25 |
|
GAD-7 |
10.00 |
12.00 |
0.116 |
4.00 |
5.00 |
0.204 |
47/25 |
|
MoCA |
18.00 |
17.00 |
0.056 |
21.00 |
21.00 |
0.397 |
47/25 |
|
WHO Physical |
48.00 |
45.00 |
0.313 |
79.00 |
64.00 |
0.041 |
47/25 |
|
WHO Psychological |
53.71 ± 15.95 |
47.60 ± 16.89 |
0.142 |
73.50 |
70.00 |
0.229 |
47/25 |
|
WHO Social |
56.00 |
56.00 |
0.944 |
70.00 |
67.00 |
0.052 |
47/25 |
|
WHO Environment |
59.98 ± 13.71 |
56.48 ± 13.48 |
0.300 |
79.44 ± 12.20 |
67.84 ± 11.73 |
0.000 |
47/25 |
Comment 3: I did not see the multivariate analyssi table for the predictors in the paper though it is mentioned in the clarifications to the reviewers comments
Answer 3: We appreciate the reviewer’s observation. As outlined in the revised Statistical Analysis section, we performed multivariable logistic regression (improved vs. not improved) and linear regression on Δ-scores (T2–T1), adjusting for age, sex, baseline score, CPAP adherence, and residual AHI for each patient-reported outcome. To focus on clinically meaningful predictors, only independent associations that reached or approached statistical significance were retained and are explicitly reported in the Results section, rather than presenting a full regression table. We added the following to the text at lines 325-328.
Additionally, we provided the data obtained during the multivariable logistic regression analysis in the supplementary file entitled „Supplementary file – Table S1”
„For each patient-reported outcome, we fit logistic models (improved vs. not improved) and linear sensitivity analyses on Δ-scores (defined so that positive scores indicate improvement), adjusting for age, sex, baseline score, CPAP adherence (% over 6 months), and residual AHI. Residual AHI did not independently predict improvement for any outcome (all p > 0.05). Consistently, higher CPAP adherence showed significant or near-significant associations with improvements in mood and quality-of-life domains in the linear models (e.g., PHQ-9, WHO-Social, WHO-Environment), whereas residual AHI did not. After adjustment, no post-treatment sex differences were observed for ESS, PHQ-9, GAD-7, or MoCA (all p > 0.20). In contrast, quality-of-life recovery favoured men in the WHOQOL-BREF Physical (p = 0.041) and Environment (p < 0.001) domains. A more negligible sex-related effect was also observed in the psychological domain (p = 0.032), but this did not persist across sensitivity models. Overall, these findings highlight CPAP adherence—rather than residual respiratory burden—as the principal modifiable driver of 6-month patient-reported benefit”
Reviewer 3 Report
Comments and Suggestions for Authors
Dear authors,
Polygraphy data from all participants should be presented as central figures.
The Intraclass Correlation Coefficient for assessing inter-rater reliability must be provided.
The full GAD-7 and PHQ-9 questionnaire instruments should be presented as supplementary files.
The conclusion must be rewritten. It is confusing.
Author Response
Dear Editor and Reviewer,
Thank you for your thoughtful suggestions. Below, we outline the revisions made to the manuscript
Adriana-Loredana Pintilie, on behalf of all co-authors
Comment 1:Polygraphy data from all participants should be presented as central figures.
Answer 1: We thank the reviewer for this valuable suggestion. In line with your recommendation, we have now generated a central composite figure illustrating baseline polygraphy parameters (AHI, ODI, and Mean nocturnal SpO₂), stratified by sex.
„Figure 1 illustrates the distribution of baseline polygraphy parameters-Apnea-Hypopnea Index (AHI), Oxygen Desaturation Index (ODI), and Mean nocturnal SpO₂- stratified by sex (M = males, F = females).”
Comment 2:The Intraclass Correlation Coefficient for assessing inter-rater reliability must be provided.
We appreciate the reviewer’s comment regarding inter-rater reliability. In our study design, polygraphy interpretation and questionnaire administration were performed once per patient by the patient's treating sleep physician as part of standard clinical workflow. Multiple raters performed no parallel or duplicated scoring, and all questionnaires used standardised self-scoring protocols with deterministic output (ESS, PHQ-9, GAD-7, MoCA, WHOQOL-BREF, DASS-21).
Therefore, the calculation of an Intraclass Correlation Coefficient (ICC) was not applicable, as ICC requires multiple raters assessing the same patient data independently under comparable conditions. We have clarified this in the Statistical analysis section with the following sentence:”Clinical polygraphy and questionnaire scoring were performed per patient by a single treating physician (polygraphy) and by self-administered instruments with deterministic scoring (questionnaires). As independent duplicate ratings were not obtained, inter-rater reliability statistics such as Interclass Correlation Coefficient (ICC) did not apply to this data set.” ( lines 177-181)
Comment 3:The full GAD-7 and PHQ-9 questionnaire instruments should be presented as supplementary files.
Answer 3: Thank you for this helpful suggestion. We have included the complete GAD-7 and PHQ-9 instruments as Supplementary Files, as they were used in our research.
Comment 4: The conclusion must be rewritten. It is confusing.
Comment 4: We have comprehensively rewritten the conclusion to adress this concern, as it follows:
‘’Beyond objective physiological parameters, OSA impairs multiple dimensions of daily functioning and societal participation. The multidomain burden documented in our cohort (excessive daytime sleepiness in 68,9%, cognitive impairment in 98,6% and mood disturbances in over 60%) translates directly into reduced occupational productivity, increased workplace absenteeism, high risk of work-related accidents, and withdrawal from social engagement. These functional impairments create a bidirectional relationship: OSA-related symptoms compromise professional and personal functioning, while unemployment and social isolation exacerbate metabolic and psychiatric comorbidities, perpetuating a vicious cycle of disability.
This single-center cohort study of adults with moderate to severe OSA revealed a significant multidomain burden at the time of diagnosis and demonstrated substantial, though incomplete, improvement after six months of CPAP therapy. Notably, one of the most affected aspects was excessive daytime sleepiness, which decreased after six months, with the ESS score dropping to 8.6 ± 3.7; however, residual excessive daytime sleepiness was still present in 27.0% of the participants. The high prevalence of cognitive impairment, assessed via the MoCA-Blind questionnaire, improved to 25.3 ± 2.7; nevertheless, it affected 48.6% of the study population. The rates of mood disturbances, evaluated through the PHQ-9 and GAD-7 scales, decreased to 28.4% and 22.9% respectively. Quality of life also improved, with the main domains of the WHOQOL-BREF increasing at six months, although a few patients scored ≤50. Higher adherence to CPAP therapy was associated with a greater reduction in ESS scores and tended to lower PHQ-9 scores, reinforcing adherence as a key modifiable factor influencing patient-centred outcomes.
From a clinical perspective, these findings reinforce adherence as one of the top targets in CPAP management and necessitate routine multidomain screening at diagnosis, patient counseling regarding expected benefits and residual symptoms, and structured reassessment protocols monitoring PRO, adherence, and residual respiratory indices. From a public health perspective, optimizing CPAP adherence through targeted interventions may reduce health care utilization, restore workforce participation, and alleviate the substantial societal burden of untreated sleep-disordered breathing. ‘’
